# A Longitudinal Study of Perceptions of the Massachusetts Menthol Ban and Its Impact on Smoking Behaviors among Marginalized Individuals

**DOI:** 10.3390/ijerph20105790

**Published:** 2023-05-11

**Authors:** Anna Booras, Renda Soylemez Wiener, Jennifer Maccarone, Andrew C. Stokes, Jessica L. Fetterman, Naomi M. Hamburg, Johar Singh, Katia Bulekova, Hasmeena Kathuria

**Affiliations:** 1The Pulmonary Center, Boston University Chobanian & Avedisian School of Medicine (BUSM), Boston, MA 02118, USA; 2Center for Healthcare Organization & Implementation Research, VA Boston Healthcare System, Boston, MA 02130, USA; 3National Center for Lung Cancer Screening, Veterans Health Administration, Washington, DC 20422, USA; 4Department of Global Health, Boston University School of Public Health, Boston, MA 02118, USA; 5Evans Department of Medicine and Whitaker Cardiovascular Institute, Boston University Chobanian & Avedisian School of Medicine (BUSM), Boston, MA 02118, USA; 6Research Computing Services, Information Services & Technology, Boston University, Boston, MA 02215, USA

**Keywords:** Massachusetts menthol ban, menthol cigarettes, tobacco-related health disparities, pre- and post-survey, qualitative study

## Abstract

Menthol cigarettes have had a profound adverse effect on public health. On 1 June 2020, Massachusetts became the first state to ban the sale of menthol cigarettes. We explored how perceptions of the ban and smoking behaviors changed over time among a group of 27 individuals who smoked menthol cigarettes at our safety-net hospital. In a convergent mixed methods study, we administered questionnaires and interviews simultaneously at two timepoints: 1 month pre-ban and 6 months post-ban. Pre-ban, we assessed perceptions of the ban and anticipated smoking behaviors after the ban. Post-ban, we assessed participants’ actual smoking behaviors and elicited suggestions to avoid unintended consequences that might undermine intended policy effects. Several respondents perceived the Massachusetts ban as positive because it could promote smoking cessation, prevent youth initiation, and mitigate unfair targeting of socioeconomically disadvantaged populations. Others perceived the ban as an overreach of government policy, financially motivated, and unfairly targeting the Black community. Many continued to smoke menthol cigarettes obtained outside Massachusetts. Individuals suggested promoting tobacco treatment for people affected by the ban and a national ban to circumvent out-of-state purchasing of menthol cigarettes. Our findings suggest that in order to be most effective, healthcare systems must promote tobacco treatment and ensure that treatment is accessible to all individuals affected by the ban.

## 1. Introduction

Menthol cigarettes have had a profound adverse effect on public health [1,2]. Menthol cigarettes increase youth initiation of smoking, make cigarettes harder to quit, and cause disproportionate death among Black Americans [3]. The US 2009 Family Smoking Prevention and Tobacco Control Act banned cigarettes with characterizing flavors, but exempted menthol [4]. Since 2011, menthol cigarette sales have increased [5]. Overall, 30% of White versus 85% of Black individuals who currently smoke use menthol cigarettes [6,7]. A recent study showed that rates of menthol cigarette use among adults who smoke cigarettes is particularly high for people who identify as lesbian or gay (51%) and bisexual (46%); people with mental health problems (45%) and past 30-day alcohol use (48%); and socioeconomically disadvantaged populations (47%) [6,7]. Targeted marketing by the tobacco industry is implicated in the disproportionate use of menthol observed among these individuals, which is most prominently evident in the Black community [8,9].

One policy solution to address the health effects of menthol cigarettes is a ban on sales. On 1 June 2020, Massachusetts became the first state to restrict the sale of all flavored tobacco products, including menthol cigarettes. In April 2021, the US Food & Drug Administration (FDA) proposed product standards to ban menthol as a characterizing flavor in cigarettes and to ban all characterizing flavors, including menthol, in cigars [10,11,12,13]. Simulation modeling to project the impact of a US menthol ban suggested that by 2060, the relative reduction in smoking prevalence would be 15.1% overall and 48.2% of Black individuals would become non-users [13,14,15,16].

Little is known about how such policies are perceived by the public, particularly individuals who are receiving treatment for tobacco dependence. Understanding how historically marginalized populations with high menthol cigarette use perceive the ban can inform tobacco treatment interventions and policy implementation of the anticipated FDA rule. We, therefore, sought to characterize how individuals cared for by Boston Medical Center (BMC), the largest urban safety-net hospital in Massachusetts, perceived the state menthol ban and how it impacted smoking behaviors over time. BMC serves a predominantly minority (70%) and low-income (50% with yearly income less than USD 20,420) population with high smoking rates (25%) [17]. Among hospitalized individuals who smoked cigarettes at BMC, the rate of menthol cigarette use the year before the Massachusetts menthol ban was 51% (410/806) overall and 68% (213/313) among Black patients, compared to 37.8% (155/421) among White patients (*p* < 0.001). We undertook a mixed methods study to explore how the Massachusetts ban affected perceptions and smoking behaviors among individuals who smoke menthol cigarettes over time, and to elicit suggestions to mitigate unintended consequences of the ban that might undermine the intended policy effects.

## 2. Materials and Methods

### 2.1. Recruitment and Enrollment

From 15 April–31 May 2020 (pre-ban), we telephoned BMC patients (*n* = 35) who currently smoked menthol cigarettes according to BMC inpatient or outpatient tobacco treatment specialist notes. The standard of care at Boston Medical Center is that all hospitalized patients, regardless of clinical condition or readiness to quit, receive inpatient tobacco counseling and recommendations for pharmacotherapy upon hospital discharge [18]. All patients enrolled in this study had at least one counseling session with a tobacco-trained specialist [18]. Eligible participants were: (1) ≥18 years old, (2) English-speaking, and (3) in agreement to participate in pre-ban and 6-month surveys/interviews. Twenty-seven individuals consented to enroll in the study and completed the pre-ban survey. Fourteen of these individuals completed the post-ban survey. We compensated individuals with up to USD 50: USD 25 for the pre-ban survey/interview and USD 25 for the 6-month post-ban survey/interview. Our Institutional Review Board approved this study.

### 2.2. Study Design and Data Collection

This mixed methods study used a convergent parallel design, with simultaneous collection of quantitative data through surveys and qualitative data through semi-structured interviews [19]. The quantitative and qualitative data collection were designed to answer related aspects of the same research questions (i.e., how perceptions of the MA menthol ban and smoking behaviors changed over time). We used a parallel mixed data analysis approach, which involved two independent processes: quantitative analysis of the survey data and qualitative analysis of interview data. Triangulation of the survey and interview data was then used to gain a more comprehensive understanding of patients’ perceptions and smoking behaviors in response to the Massachusetts menthol ban [20].

#### Surveys and Qualitative Interviews

Two research team members with no clinical relationships with participants conducted the baseline and six-month post-ban surveys and semi-structured interviews; questionnaires and interviews were administered during the same telephone encounter. Responses were audio-recorded and transcribed. Pre-ban questionnaires included demographics and cigarette use characteristics. To compare how smoking behaviors and perceptions changed over time, pre-ban and 6-month post-ban questionnaires assessed the purchasing and use of tobacco products. The pre-ban semi-structured interview guide was designed to identify participants’ perceptions of the Massachusetts menthol ban, elicit their opinions on why the ban was enacted, and probe their anticipated smoking behaviors after the ban went into effect. The 6-month post-ban interview guides were designed to probe how perceptions and smoking behaviors changed over time and to elicit suggestions on how to strengthen the ban.

### 2.3. Data Analyses

The pre-ban and 6 months post-ban questionnaire responses are summarized (frequency and percentages) in the tables as follows. Table 1: Baseline characteristics of participants; Table 2: Participants’ perceptions of the ban on attitudes and menthol cigarette sales (pre- and 6 months post-ban); Table 3: Perceived influence on smoking behavior (6 months post-ban); and Table 4: Purchasing and use of tobacco products (pre- and 6 months post-ban). In addition, paired *t*-tests were used to compare mean cigarettes smoked per day at the pre-ban versus 6 months post-ban timepoints.

For the qualitative analysis, we analyzed transcripts using inductive content analysis [23,24] by performing unstructured coding to allow for the identification of new themes. Two team members reviewed the first several transcripts independently and developed a preliminary coding matrix. The team members then manually coded all transcripts in full. Through constant comparison and iterative discussion, codes were revised and added until a consensus on summary categories was reached. We finalized categories, grouped themes in each category, and identified quotes that best highlighted individual themes. Supporting statements are identified by participant number.

## 3. Results

Table 1 shows the baseline demographics and smoking characteristics of the study participants. Participants came from historically marginalized groups: 55% were of the Black race, 19% of Hispanic ethnicity, 92% had a household income <$35,000, and 41% had not completed high school. Participants on average had smoked for a mean of 33.4 ± 14.5 years, and 55% (15/27) smoked 10 or more cigarettes a day. Eighty-five percent (23/27) smoked Newport menthol cigarettes.

### 3.1. Pre-Ban Opinions on Why the Ban Was Enacted

In pre-ban interviews (*n* = 27), we explored participants’ opinions on why the menthol ban was enacted. We present representative quotes, identified by participant number, that best illustrate the themes captured from analysis of qualitative data. The perceived rationale for the menthol ban fell into three themes:(1)General interest in promoting health: “*It might be because they don’t want the people in Massachusetts to be smoking like they do*” (P25);(2)Financial motivation: “*I don’t really think it’s to save people. It’s probably about money. …Maybe there’s somebody who’s got their money in the non-menthol cigarettes, and they want all the game on non-menthol cigarettes*” (P23)(3)Intent to reduce the youth smoking epidemic: “*Mostly, I think, it’s for the younger people*” (P15).

### 3.2. Quantitative and Qualitative Perceptions of the Menthol Ban over Time

Fifty-two percent (14/27) completed both pre-ban and six-month assessments; these fourteen participants were included in the pre- versus post-ban comparisons presented in the remainder of the results section. Table 2 compares pre- and post-ban perceptions of the ban regarding attitudes and menthol cigarettes sales.

#### 3.2.1. Pre-Ban Perceptions

Pre-ban, 29% (4/14) participants strongly agreed or agreed with the Massachusetts menthol ban (Table 2), citing reasons that fell into three themes:(1)Ban would help individuals to stop smoking: “*I will not be smoking them because I only smoke menthol. …Soon as June 1st come, I’m done. I’m not spending my money on something I don’t like*” (P15)(2)Ban would benefit the community: “*The young people smoke Newports. …[post-ban] they won’t be able to get ’em. I think that will help not only the young people, but the seniors as well as the middle age*” (P25)(3)Ban would undo an injustice to the Black community: “*It irritates me that I can’t buy a pack [of menthol cigarettes], but I know that they’ve been marketed, unfortunately, for people of color, so it’s possible that injustice will no longer be happening. It’s a good thing*” (P26).

By contrast, other participants initially had a negative perception of the menthol ban. Two themes emerged:(1)Some believed the ban to be an overreach of government policy: “*I think they should leave the menthol cigarettes alone… They’re trying to delegitimize cigarettes in all. I don’t agree with it. I think it’s an overreach of government*” (P24).(2)Others thought the ban unfairly targeted Black communities by limiting choices for products they enjoy: “*There’s certain type of alcohols that African Americans drink that are being taken off the shelf also. The African Americans smoke menthols more than anyone else. …That’s where I see discrimination*” (P6).

Regardless of whether they perceived the ban positively or negatively, individuals were skeptical that the ban would be an effective strategy to support smoking cessation. Four themes emerged as to why participants thought the ban might be ineffective:(1)Unsanctioned re-selling of menthol cigarettes: “*It’s not gonna stop people having [menthol] cigarettes. They’re gonna just go to New Hampshire, get [menthol] cigarettes and sell ‘em by the pack, by the carton*” (P4)(2)Switching from menthol to non-menthol cigarettes: “*If you go to the store, and they don’t got your brand, you go buy what they got on the shelf—cause most people who are addicted, they gonna smoke*” (P22)(3)Traveling out of state to purchase menthol cigarettes: “*I would drive and get me a couple of cartons then come home…I would take a ride if I wanted the cigarettes*” (P8)(4)Manipulating substances to make a menthol cigarette: “*people [will] figure out some way [of] smoking their menthol cigarettes. They buy some menthol products and put ’em on some non-menthol cigarettes and turn them into menthol cigarettes. Everybody come up with something*” (P19).

#### 3.2.2. Post-Ban Perceptions

At 6 months, 36% (5/14) strongly agreed or agreed with the Massachusetts menthol ban (Table 2), commenting “*I’m glad they did it ‘cause sometimes, you need a jolt to stop something*.” (P25). Post-ban, 57% (8/14) strongly disagreed or disagreed with the Massachusetts ban: “*I hate it. …It’s harder to get menthol cigarettes. It costs more money to buy cigarettes*” (P17). Compared to pre-ban, twice as many participants reported that the ban positively impacted them (3/14, 21% pre-ban vs. 6/14, 43% post-ban).

### 3.3. Quantitative Comparisons of Smoking Behaviors and Purchase and Use of Tobacco Products Pre- vs. Post-Ban

Table 3 shows the reported changes in smoking behaviors: 57% percent (8/14) thought that the ban had encouraged them to stop using all tobacco products, and half (7/14) reported that they had made at least one quitting attempt due to the ban. Two individuals reported stopping smoking due to the ban. While the number of individuals who smoked cigarettes daily was similar pre- vs. post-ban (9/14 (64.3%) pre-ban vs. 8/14 (57%) 6 months post-ban), the mean number of cigarettes smoked per day significantly decreased (mean (SD): 13.9 (10.8) pre-ban vs. 5.8 (3.4) 6-months post-ban, *p* = 0.01).

Table 4 shows pre-ban and 6-month comparisons of the purchasing and use of non-menthol cigarettes, menthol cigarettes, and other tobacco products. Pre-ban, 71% (10/14) of individuals smoked menthol cigarettes exclusively and 29% (4/10) smoked both menthol and non-menthol cigarettes. At 6 months post-ban, only 2 of the 10 individuals who exclusively smoked menthol cigarettes pre-ban started to smoke non-menthol cigarettes. Of the 4 individuals who smoked both menthol and non-menthol cigarettes pre-ban, at 6 months post-ban 2 smoked fewer non-menthol cigarettes, 1 had no change, and 1 smoked more non-menthol cigarettes. Compared to pre-ban, 71% (10/14) reported purchasing and smoking fewer menthol cigarettes at 6 months. Most participants who continued to smoke menthol cigarettes post-ban began purchasing their cigarettes out of state: Pre-ban, 78.5% (11/14) bought menthol cigarettes in Massachusetts, while 6 months post-ban, 78.5% (11/14) bought menthol cigarettes out of state. Pre-ban, none of the 14 individuals purchased and used other tobacco products; at 6 months, one individual started to purchase and use e-cigarettes.

### 3.4. Narratives of the Impact of the Menthol Ban: Anticipated Impact and Attitudes Pre-Ban Versus Actual Behaviors and Attitudes 6 Months Post-Ban

Pre-ban, we asked participants to anticipate how they thought the menthol ban would impact their smoking behaviors. We categorized participants’ responses into three groups: (1) participants who thought the ban would lead them to stop smoking; (2) participants who thought they would still smoke post-ban; and (3) participants who were unsure of how the ban would impact their smoking behavior. We depict pre-ban anticipated smoking behaviors versus actual smoking behaviors post-ban as a Sankey diagram (Figure 1), with representative quotes from each participant category to illustrate participant narratives of the anticipated versus actual impact of the menthol ban. Pre-ban, six participants anticipated that they would stop smoking or smoke less after the ban; in fact, six months post-ban, eight participants had stopping smoking or were smoking less.

### 3.5. Participant Suggestions

Participants discussed ways to make the ban more effective, with responses falling into three main themes.

(1) Some believed that all cigarettes should be banned: “*Why not ban all cigarettes? … Just do it across the board*” (P1). One individual commented: “*I didn’t think they could honestly be distinguishing between the two like, ‘It’s okay to smoke regular cigarettes, but not something that has menthol added to it’*” (P2).

These individuals believed that a ban on all cigarettes would be more effective to achieve complete smoking cessation: “*If there was no more other cigarettes, I would stop. It would’ve been awesome, but they gave us a second choice (to switch to non-menthol cigarettes*)” (P18).

(2) Participants believed that a menthol ban could be more effective if accompanied by policies to ensure that tobacco treatment interventions were widely available to affected individuals: “*If people are addicted to something, you have to go get help to stop doing it. While putting the ban out there, you need to have a little bit more advertisement… ‘We have these places to help stop smoking. We can help you through this’*” (P4).

(3) Some believed the menthol ban should be nationwide to prevent work-arounds such as out-of-state travel to purchase menthol cigarettes: “*If you gonna ban it, you gotta ban it anywhere…cause people really traveling to get it*” (P16).

## 4. Discussion

We undertook a small exploratory study to understand how the Massachusetts ban affected the perceptions and smoking behaviors over time among patients at our safety-net hospital who smoked menthol cigarettes. We found that individuals held varied views as to why the Massachusetts menthol ban was enacted, but few perceived that it was to promote their own health. Many discussed how they smoked less because of the ban. Our pre- and post-ban analyses showed that individuals’ smoking behavior was often influenced by their ability to obtain menthol cigarettes. Individuals provided suggestions on how to make the ban more effective.

Following Ontario’s ban on menthol cigarettes, individuals who smoked menthol cigarettes had increased quitting attempts and quitting success compared to those who smoked non-menthol cigarettes after one year, with one study finding that 31% believed the menthol ban drove their smoking cessation [25,26]. We found that 50% of individuals answering 6-month follow-up questions reported that the ban led them to make a quit attempt, but few were successful at quitting smoking altogether, as many either continued smoking menthol cigarettes obtained out of state or converted to smoking non-menthol cigarettes. Nonetheless, post-ban, participants were smoking fewer cigarettes per day on average (13.9 (SD 10.8) pre-ban versus 5.8 (SD 3.4) 6 months post-ban, *p* = 0.01).

Several individuals perceived the Massachusetts menthol ban as positive because it could, in theory, promote smoking cessation, prevent youth initiation, and mitigate unfair targeting of socioeconomically disadvantaged populations. Others perceived that the ban was an overreach of government policy; financially motivated; and unfairly targeting the Black community, who enjoyed menthol cigarettes. Many individuals who initially held a negative view of the ban developed a more positive opinion over the 6 months, largely due to individuals recognizing that the ban facilitated cessation.

While participants had generally favorable views of the menthol ban, its impact on smoking behaviors was modest, perhaps due to the small sample size with large confidence ratios. Participants believed that the ban was rendered less effective in promoting cessation because of (1) unsanctioned selling of menthol cigarettes, (2) switching from menthol to non-menthol cigarettes, and (3) purchasing menthol cigarettes in nearby states. Many could not understand why only a menthol ban, and not a total cigarette ban, took effect. Increasing campaigns that educate individuals on the harms of all tobacco products are critical so that individuals do not misinterpret that non-menthol tobacco products are safe.

In a cross-sectional survey assessing the behavioral intentions of menthol smokers in the event of a national ban, nearly 40% of menthol smokers said they would quit smoking altogether if menthol tobacco products were no longer available [27]. In our study, several individuals acknowledged that a national menthol ban was needed to effectively promote smoking cessation, though some discussed how even a national ban would not prevent companies or individuals from finding a way to add menthol additives to cigarettes. Some tobacco companies began marketing flavored tobacco accessories and filter modifications after such flavors were banned from cigarettes themselves [13,28]. Monitoring the industry for such activity will be critical to avoid perpetuating tobacco-related health disparities. 

When implementing the menthol ban, individuals discussed the need to ensure that tobacco treatment programs are accessible. D’Silva and colleagues showed that the implementation of a nationwide menthol ban would require expansion of tobacco treatment programs to meet the needs of the increased number of individuals who would make a quitting attempt [29]. Expanding funding for comprehensive tobacco prevention and control agencies may be necessary in conjunction with a national menthol ban. We anticipate that successful implementation will require increased collaborations with the communities impacted by the ban, partnerships with medical professionals and advocacy groups to promote access to evidence-based tobacco cessation treatment, and mass media campaigns to raise awareness of the health effects of menthol-flavored tobacco, particularly to Black Americans and other socioeconomically disadvantaged groups.

A recent analysis of the nationally representative, cross-sectional Behavioral Risk Factor Surveillance System showed the MA ban to be associated with a 1% decline in current cigarette smoking rates in Massachusetts in contrast to comparison states. The study showed a greater reduction in current cigarette smoking among males and Hispanic females, but an increase among Black females [30]. Synthesis of data from multiple sources, including both quantitative and qualitative data, can play an important role in generating an improved understanding of how to maximize the impact of the anticipated FDA rule.

Mixed-method, triangulated designs have been used to support the design and implementation of tobacco interventions in other settings, such as for pregnant women with substance use disorders and smoke-free policies in prisons [31,32]. To our knowledge, our study is the first to use these methods to understand the perceptions of the Massachusetts menthol ban and its impact on smoking behaviors. Our convergent parallel mixed methods study, utilizing a triangulation approach for data analysis, allowed us to make complementary inferences of the quantitative and qualitative data. Drawing on the combined strengths of quantitative and qualitative data allowed us to gain a richer understanding of our results, contextualized participants’ perceptions of the menthol ban and smoking behaviors, lessened bias associated with single-method approaches, and allowed for a more robust analysis, despite our small study size.

Another strength of our study is that it was conducted in a historically marginalized population, in a setting with a high prevalence of menthol cigarette use. On the other hand, our small sample size from a single recruitment site may limit generalizability. Perceptions of participants recruited from a hospital setting may differ in significant ways from the general population. Future studies should include a larger sample size from diverse settings. Our study is unique in that it followed patients longitudinally from pre-ban to 6 months post-ban. While the sample size was small, the longitudinal qualitative study provided a rich understanding of the lived experiences and perspectives of participants impacted by the menthol ban. Though retention was low, we achieved our goal to collect data from about 15 patients at 6 months post-ban; given the likelihood of attrition in socioeconomically disadvantaged populations, we had purposefully aimed to recruit 30 participants for the study. Modeling of predictive reactions from the menthol ban suggests that the reaction period could last longer than 1 year; future studies should follow patients for at least this long [15].

## 5. Conclusions

Finalizing and implementing the FDA rule to ban menthol as a characterizing flavor in cigarettes, and to ban all characterizing flavors, including menthol, in cigars, is a critical step to reducing tobacco-related health disparities and promoting health equity [1]. Healthcare workers must capitalize on this moment to promote tobacco treatment, and states must ensure that tobacco treatment programs are accessible to support individuals affected by the ban. Given the previous injustice of the tobacco industry to the Black and low-socioeconomic-class communities, it is critical to ensure increased access to culturally tailored smoking cessation services and evidence-based smoking cessation interventions that are barrier-free and covered by all health insurance types.

## Figures and Tables

**Figure 1 ijerph-20-05790-f001:**
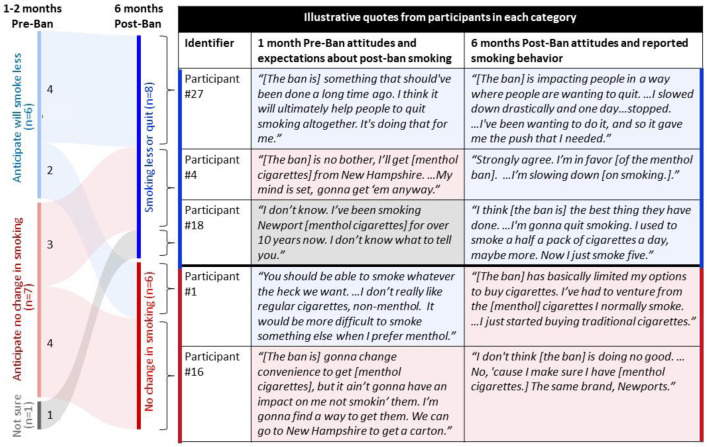
Comparison of anticipated and actual smoking behavior post-ban, depicted by a flow diagram. Anticipated and actual smoking behavior 6 months post-ban with representative quotes are depicted in the flow diagram and table. Following the diagram from left to right, 6 participants anticipated they would stop smoking or smoke less post-ban (blue); following the blue arrows to 6 months post-ban, 4 participants stopped smoking or smoked less (blue solid bar) and 2 participants did not change their smoking behavior. Seven participants anticipated that they would still smoke post-ban (red); following the red arrows to six months post-ban, three participants stopped smoking or smoked less (red solid bar) and four participants did not change their smoking behavior. One participant was unsure of how the ban would affect their smoking behavior; following the grey arrow to six months post-ban, the participant stopped smoking or smoked less.

**Table 1 ijerph-20-05790-t001:** Characteristics of individuals participating in the study.

Characteristics	Baseline Assessments (*n* = 27)
Mean Age (SD)	52 (12.4)
Female	10 (37%)
Race	
Black	15 (55%)
White	8 (30%)
Other	3 (11%)
Mixed race (2 or more races)	1 (4%)
Hispanic Ethnicity	5 (19%)
Medicaid Insurance	26 (96%)
Education	
Less than high school	11 (41%)
High school/GED	7 (26%)
More than high school	8 (30%)
Experiencing homelessness	9 (33%)
Unmarried *	24 (89%)
Unemployed	26 (96%)
Yearly household income before taxes	
USD 0–34,999	25 (92%)
>USD 35,000	1 (4%)
Prefer not to answer/Don’t know	1 (4%)
Depression and/or anxiety	17 (63%)
Current use of substances	
Alcohol (5+ men, 4+ women in 1 day)	9 (33%)
Cocaine	7 (26%)
Opioids	4 (15%)
Marijuana	5 (19%)
Cigarette smoking characteristics	
Years smoked (SD)	33 (14.5)
Smokes daily	19 (70%)
Smokes 10 or more cigarettes a day	15 (55%)
Mean Fagerstrom score (SD) **	4.4 (2.5)
Menthol use exclusively	20 (74%)
Current use of other tobacco products	
Electronic cigarettes	1 (4%)
Cigars	1 (4%)
No other tobacco product use	25 (92.5%)

* Unmarried comprises divorced/separated, widowed, and never married. ** Fagerstrom Test for Cigarette Dependence [21,22]. Scores range from 0 to 10. No dependence corresponds to a score of 0, low dependence a score of 1 or 2, low-to-moderate dependence a score of 3 or 4, moderate dependence a score of 5 to 7, and high dependence a score of 8 to 10.

**Table 2 ijerph-20-05790-t002:** Perceptions of the ban in terms of attitudes and menthol cigarette sales in Massachusetts and nationwide.

Perceptions of Ban	Pre-Ban Perceptions (*n* = 14)	6 m Post-Ban Perceptions (*n* = 14)
In Favor of the Ban on Menthol Cigarette Sales in Massachusetts		
Strongly agree	1 (7%)	5 (36%)
Agree	3 (21%)	0 (0%)
Neither agree nor disagree	3 (21%)	1 (7%)
Disagree	3 (21%)	6 (43%)
Strongly disagree	4 (29%)	2 (14%)
In Favor of a Ban on Menthol Cigarette Sales Nationwide		
Strongly agree	1 (7%)	5 (36%)
Agree	3 (21%)	1 (7%)
Neither agree nor disagree	3 (21%)	0 (0%)
Disagree	3 (21%)	6 (43%)
Strongly disagree	4 (29%)	2 (14%)
Anticipated/Perceived Impact of the Menthol Ban on Participant		
Negatively impact me	1 (7%)	2 (14%)
Not impact me at all	8 (57%)	6 (43%)
Positively impact me	3 (21%)	6 (43%)
Did not answer/not sure	2 (14%)	-

**Table 3 ijerph-20-05790-t003:** Perceived influence on smoking behavior of the menthol ban at 6 months.

Perceived Influence	6 m Post-Ban Behavior (*n* = 14)
The ban encouraged me to stop using all tobacco products	
Strongly agree/agree	8 (57%)
Neither agree nor disagree	1 (7%)
Strongly disagree/disagree	5 (34%)
The ban made me want to stop using all cigarettes	
Strongly agree/agree	5 (34%)
Neither agree nor disagree	2 (14%)
Strongly disagree/disagree	7 (50%)
The menthol ban has led me to make a quit attempt	
Yes	7 (50%)
No	7 (50%)
The menthol ban has led me to stop smoking	
Yes	2 (14%)
No	12 (86%)

**Table 4 ijerph-20-05790-t004:** Pre-ban and 6-month comparisons of the purchasing and use of tobacco products.

Impact of Ban on Purchase and Use of Menthol Cigarettes, Non-Menthol Cigarettes and Other Tobacco Products		
	Anticipated Behavior Pre-Ban (*n* = 14)	Actual Behavior 6 m Post-Ban (*n* = 14)
Purchase of non-menthol cigarettes		
Participants who purchase non-menthol and menthol		
Stop buying non-menthol cigarettes	0 (0%)	0 (0%)
Buy fewer non-menthol cigarettes	0 (0%)	1 (7%)
No change in how many non-menthol cigarettes I buy	2 (14%)	3 (21%)
Buy more non-menthol cigarettes	2 (14%)	0 (0%)
Participants who purchase exclusively menthol cigarettes		
Start buying non-menthol cigarettes	1 (7%)	3 (21%)
Did not buy before and will not buy after	7 (50%)	7 (50%)
Did not answer/not sure	2 (14%)	0 (0%)
Non-menthol cigarette use		
Participants who smoke both non-menthol and menthol		
Stop smoking non-menthol cigarettes	0 (0%)	0 (0%)
Smoke non-menthol cigarettes less	0 (0%)	2 (14%)
No effect on how much I smoke non-menthol cigarettes	3 (21%)	1 (7%)
Smoke non-menthol cigarettes more	1 (7%)	1(7%)
Participants who smoke exclusively menthol cigarettes		
Start smoking non-menthol cigarettes	1 (7%)	2 (14%)
Continue to refrain from non-menthol cigarettes	7 (50%)	8 (57%)
Did not answer/not sure	2 (14%)	0 (0%)
Purchase of menthol cigarettes		
Stop buying menthol cigarettes	5 (36%)	4 (29%)
Buy fewer menthol cigarettes	2 (14%)	6 (43%)
No change in how many menthol cigarettes I buy	4 (29%)	3 (21%)
Buy more menthol cigarettes	0 (0%)	1 (7%)
Did not answer/not sure	3 (21%)	0 (0%)
Menthol cigarette use		
Stop smoking menthol cigarettes	5 (36%)	4 (29%)
Smoke menthol cigarettes less	2 (14%)	6 (43%)
No effect on how much I smoke menthol cigarettes	4 (29%)	3 (21%)
Smoke menthol cigarettes more	0 (0%)	1 (7%)
Did not answer/not sure	3 (21%)	0 (0%)
Purchase of other tobacco products		
Buy more other tobacco products	2 (14%)	0 (0%)
Start buying other tobacco products	0 (0%)	1 (7%)
Continue to refrain from other tobacco products	12 (85.7%)	13 (93%)
Tobacco product use		
Smoke other tobacco products more	1 (7%)	0 (0%)
Start using other tobacco products	1 (7%)	1 (7%)
Continue to refrain from other tobacco products	12 (85.7%)	13 (93%)
Impact of ban on purchasing behaviors of menthol cigarettes		
	Pre-ban behavior	6 m post-ban behavior
State where menthol cigarettes are bought		
Massachusetts	11 (79%)	3 (21%)
Other State	2 (14%)	11 (79%)
Missing data	1 (7%)	-

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
