# Peer review of "A Longitudinal Study of Perceptions of the Massachusetts Menthol Ban and Its Impact on Smoking Behaviors among Marginalized Individuals"

_ijerph, 2023, doi:10.3390/ijerph20105790_

Round 1

Reviewer 1 Report

A Longitudinal Study of Perceptions of the Massachusetts Menthol Ban and Its Impact on Smoking Behaviors among Marginalized Individuals

Thank you for the opportunity to review this manuscript, which examined individuals’ perceptions of a ban on the sale of menthol cigarettes in Massachusetts. Participants were assessed 1-month pre-pan and 6-months post-ban to examine their anticipated smoking behaviors after the ban and participants’ actual smoking behaviors after the ban, as well as their suggestions to avoid unintended consequences of the ban. This is an interesting study with important implications for public health and health equity. I have a few minor comments to strengthen the manuscript.

Introduction:

1.       I suggest providing rates of menthol use among the minoritized populations mentioned in the introduction (i.e., sexual minority individuals, those with mental health problems, and socioeconomically disadvantaged populations) to strengthen the introduction.

2.       In the last paragraph (first sentence), the authors state that little is known about how these policies are perceived by patients. This seems to come out of nowhere given no mention of patients prior to this paragraph. I recommend making this sentence a bit broader to guide the reader toward the sample used for the current study – for instance, “Little is known about how such policies are perceived by the public, particularly individuals who are seeking treatment for tobacco cessation.”

Methods:

1.       It would be helpful to note how many potential patients were contacted to participate in the study to get some sense of the proportion of those who agreed to participate among those were contacted.

2.       Did all 27 participants complete both the pre-ban and post-ban survey/interview? Did anyone complete only pre-ban?

3.       Do the authors have any kappas to report to provide a sense of inter-rater reliability on the qualitative coding?

4.       Were study participants receiving any tobacco treatment prior to participating in this study? I don’t think they were, but it’s a bit unclear given the mention of inpatient/outpatient tobacco treatment specialist notes. I recommend clearly stating this.

Results:

1.       Do the authors have information on participants’ sexual identity? This seems particularly important given that this was mentioned as a risk factor for menthol smoking in the introduction. Same for mental health symptoms/diagnoses.

2.       Is the information provided under smoking characteristics in Table 1 focusing solely on cigarette smoking? Did the authors assess information on the use of other tobacco products?

3.       I now see that the authors provide information on the number of participants who completed both pre- and post-ban assessments. I might consider presenting this earlier on in the methods so that readers are not left wondering.

Discussion:

1.       Overall, I thought the discussion was well-written, adequately summarized research findings, and noted implications of findings.

Author Response

Reviewer: 1

General Comments to Author
Thank you for the opportunity to review this manuscript, which examined individuals’ perceptions of a ban on the sale of menthol cigarettes in Massachusetts. Participants were assessed 1-month pre-pan and 6-months post-ban to examine their anticipated smoking behaviors after the ban and participants’ actual smoking behaviors after the ban, as well as their suggestions to avoid unintended consequences of the ban. This is an interesting study with important implications for public health and health equity. I have a few minor comments to strengthen the manuscript.

Response: Thank you for the positive feedback and your suggestions to strengthen the manuscript. We have responded to your questions and comments below.

Specific Comments to Author

Introduction:

  1. I suggest providing rates of menthol use among the minoritized populations mentioned in the introduction (i.e., sexual minority individuals, those with mental health problems, and socioeconomically disadvantaged populations) to strengthen the introduction.

Response: Thank you for this suggestion. We added (Page 1-2, lines 42-46): A recent study showed that rates of menthol cigarette use among adults who smoke cigarettes is particularly high for people who identify as lesbian or gay (51%) and bisexual (46%); people with mental health problems (45%) and past 30-day alcohol use (48%); and socioeconomically disadvantaged populations (47%).(6, 7) 

  1. In the last paragraph (first sentence), the authors state that little is known about how these policies are perceived by patients. This seems to come out of nowhere given no mention of patients prior to this paragraph. I recommend making this sentence a bit broader to guide the reader toward the sample used for the current study – for instance, “Little is known about how such policies are perceived by the public, particularly individuals who are seeking treatment for tobacco cessation.”

Response:  We have modified the text to read (Page 2, lines 57-58): Little is known about how such policies are perceived by the public, particularly individuals who are receiving treatment for tobacco dependence.

Methods:

  1. It would be helpful to note how many potential patients were contacted to participate in the study to get some sense of the proportion of those who agreed to participate among those were contacted.

Response: We added (Page 2, Line 74): From April 15-May 31, 2020 (pre-ban), we telephoned BMC patients (n=35) who currently smoked menthol cigarettes according to BMC inpatient or outpatient tobacco treatment specialist notes.

  1. Did all 27 participants complete both the pre-ban and post-ban survey/interview? Did anyone complete only pre-ban?

Response: As the reviewer has pointed out (Results, Comment 3), we had previously written (Page 4, Lines 138-140) Fifty-two percent (14/27) completed both pre-ban and 6-month assessments; these 14 participants were included in the pre- versus post-ban comparisons presented in the remainder of the results section.

We have now additionally added in the Methods (Page 2, Line 81-83): Twenty-seven individuals consented to enroll in the study and completed the pre-ban survey. Fourteen of these 27 individuals completed the post-ban survey.

  1. Do the authors have any kappas to report to provide a sense of inter-rater reliability on the qualitative coding?

Response: Thank you for this suggestion that we will include in future studies. We do not have kappas to report.

We provide further detail on our approach to analysis of qualitative data (Page 3, Lines 109-115): Two team members reviewed the first several transcripts independently and developed a preliminary coding matrix. The team members then manually coded all transcripts in full. Through constant comparison and iterative discussion, codes were revised and added until consensus on summary categories was reached. We finalized categories, grouped themes in each category, and identified quotes that best highlighted individual themes. Supporting statements are identified by participant number. 

  1. Were study participants receiving any tobacco treatment prior to participating in this study? I don’t think they were, but it’s a bit unclear given the mention of inpatient/outpatient tobacco treatment specialist notes. I recommend clearly stating this.

Response: We now more clearly state that all patients enrolled in the study had participated in at least one counseling session with a tobacco trained specialist.

We added (Page 2, Lines 76-79): The standard of care at Boston Medical Center is that all hospitalized patients, regardless of clinical condition or readiness to quit, receive inpatient tobacco counseling and recommendations for pharmacotherapy at hospital discharge. All patients enrolled in this study had at least one counseling session with a tobacco-trained specialist.

Results:

  1. Do the authors have information on participants’ sexual identity? This seems particularly important given that this was mentioned as a risk factor for menthol smoking in the introduction. Same for mental health symptoms/diagnoses.

Response:

We have added information on mental health diagnoses and substance use in Table 1. We agree that it is important to collect information on sexual identity. We did not collect this information. We will add this measure to our surveys in future studies.

  1. Is the information provided under smoking characteristics in Table 1 focusing solely on cigarette smoking? Did the authors assess information on the use of other tobacco products?

Response: We assessed information on other tobacco product use. We have added this to Table 1. As shown in Table 1, the majority of patients (93%; 25/27) did not use other tobacco products.

  1. I now see that the authors provide information on the number of participants who completed both pre- and post-ban assessments. I might consider presenting this earlier on in the methods so that readers are not left wondering.

Response: We agree that we should present this information earlier in the methods.

We added (Page 2, Line 81-83): Twenty-seven individuals consented to enroll in the study and completed the pre-ban survey. Fourteen of these 27 individuals completed the post-ban survey

Discussion:

  1. Overall, I thought the discussion was well-written, adequately summarized research findings, and noted implications of findings.

Response: We appreciate your positive feedback.

Reviewer 2 Report

This study aimed to study how perceptions of the Massachusetts menthol ban and smoking behaviors changed over time among a group of 27 individuals smoked menthol cigarettes at their safety-net hospital by using questionnaires. The results indicated that several perceived the Massachusetts ban promoted smoking cessation, prevent youth initiation, and mitigate unfair targeting of socioeconomically disadvantaged populations. Others perceived the ban as an overreach of government policy, financially motivated, and unfairly targeting the Black community. Many people continued to smoke menthol cigarettes obtained outside Massachusetts.

1.         The confounding factors of the participant in this study did not describe and exclusive in the analysis. For example, the selection bias of the participants.

2.         The factors related to smoking cessation or change of smoking behaviors were complex. The study design of this work was imperfectly.

3.         This study presented some information and influence about the ban of menthol cigarettes; however, the sample size was only 27. In addition, after 6 months of the ban, only 14 people was ivolved. The number size was too small to represent the real results of the ban of Massachusetts menthol cigarettes. The change of smoking behavior of people outside the safety-net hospital must involve in this study.

4.         The survey and qualitative interviews were done pre-ban and 6 months after ban. How the time de In Table 1, the total and percentage of race was not meet to the sample size. In addition. the format of data must be consistent.

5.         The meaning and impact of qualitative interviews from the participant in this survey must describe and discuss deeply, not only present the answers.

6.          The format of all the reference must follow the format of Int. J. Environ. Res. Public Health.

Author Response

Reviewer: 2

Specific Comments to Author

  1. The confounding factors of the participant in this study did not describe and exclusive in the analysis. For example, the selection bias of the participants.

Response: Thank you for your feedback. We acknowledge that participants in this study, namely patients treated at Boston Medical Center who have been seen by a tobacco treatment specialist, may differ from the more general population of people who smoke menthol cigarettes but are not recruited from the hospital setting or clinics. Due to this, our sample and the target population may differ in significant ways, limiting generalizability. In addition, participants who completed both the pre-ban and 6-month post-ban assessments may have had more favorable perceptions of the MA menthol ban.

To address this, we have expanded the Limitations section as follows (Page 11, Lines 308-320):  A strength of our study is that it was conducted in a historically marginalized population, in a setting with a high prevalence of menthol cigarette use. On the other hand, our small sample size from a single recruitment site may limit generalizability. Perceptions of participants recruited from a hospital setting may differ in significant ways from the general population. Future studies should include a larger sample size from diverse settings. Our study is unique in following patients longitudinally from pre-ban to 6 months post-ban. While the sample size is small, the longitudinal qualitative study provided a rich understanding of the lived experiences and perspectives of participants impacted by the menthol ban. Though retention was low, we achieved our goal to collect data for about 15 patients at 6 months post-ban; given likely attrition in socioeconomically disadvantaged populations, we had purposively aimed to recruit 30 participants for the study. Modeling of predictive reactions from the menthol ban suggests that the reaction period could last longer than 1 year; future studies should follow patients for at least this long.(15)

  1. The factors related to smoking cessation or change of smoking behaviors were complex. The study design of this work was imperfectly.

Response: We agree that factors related to smoking cessation and change in smoking behaviors are complex. We therefore designed as study that used both quantitative and qualitative data and that analyzed perceptions and behaviors longitudinally. To our knowledge, there are few existing published manuscripts that have sought to analyze and compare qualitative results over time.  These complex factors should be further explored in future studies that address the limitations we outlined above (e.g. larger studies, recruitment from multiple sites). 

In the Limitations section (Page 11, Lines 312-320), we have also added: Future studies should include a larger sample size from diverse settings. … Modeling of predictive reactions from the menthol ban suggests that the reaction period could last longer than 1 year; future studies should follow patients for at least this long.(15)

  1. This study presented some information and influence about the ban of menthol cigarettes; however, the sample size was only 27. In addition, after 6 months of the ban, only 14 people was involved. The number size was too small to represent the real results of the ban of Massachusetts menthol cigarettes. The change of smoking behavior of people outside the safety-net hospital must involve in this study.

Response: Thank you for your feedback. We agree that future studies should include recruitment from multiple sites and larger sample sizes. Studies that use both quantitative (e.g., large data sets) and qualitative methods are complimentary and can provide a more in-depth understanding of the impact of the menthol ban. We have addressed this feedback as follows:

(1) We acknowledge in the first sentence of the Discussion section that this is a small exploratory study from one institution (Page 10, Lines 254-256): We undertook a small exploratory study to understand how the Massachusetts ban affected perceptions and smoking behaviors over time among patients at our safety-net hospital who smoke menthol cigarettes.

(2) We further acknowledged the limitations of our study (Page 11, Lines 308-320):  A strength of our study is that it was conducted in a historically marginalized population, in a setting with a high prevalence of menthol cigarette use. On the other hand, our small sample size from a single recruitment site may limit generalizability. Perceptions of participants recruited from a hospital setting may differ in significant ways from the general population. Future studies should include a larger sample size from diverse settings. Our study is unique in following patients longitudinally from pre-ban to 6 months post-ban. While the sample size is small, the longitudinal qualitative study provided a rich understanding of the lived experiences and perspectives of participants impacted by the menthol ban. Though retention was low, we achieved our goal to collect data for about 15 patients at 6 months post-ban; given likely attrition in socioeconomically disadvantaged populations, we had purposively aimed to recruit 30 participants for the study. Modeling of predictive reactions from the menthol ban suggests that the reaction period could last longer than 1 year; future studies should follow patients for at least this long.(15)

(3) We added text on how quantitative and qualitative studies can work together to maximize the impact of the anticipated FDA rule and included a recent study that conducted a cross-sectional analysis of the MA menthol ban (Page 11, 301-307): A recent analysis of the nationally representative, cross-sectional Behavioral Risk Factor Surveillance System showed the MA ban to be associated with a 1% decline in current cigarette smoking rates in Massachusetts than comparison states. The study showed a greater reduction in current cigarette smoking among males and Hispanic females while increasing among Black females.(24) Synthesis of data from multiple sources including both quantitative and qualitative data can play an important role in generating an improved understanding of how to maximize the impact of the anticipated FDA rule.

  1. The survey and qualitative interviews were done pre-ban and 6 months after ban. How the time de In Table 1, the total and percentage of race was not meet to the sample size. In addition. the format of data must be consistent.

Response: Thank you for your feedback.

In the demographic survey portion of the study, we allowed participants (N=27) to select more than one race. One participant selected two race designations, which is why the total number participants within the race category adds up to 28. To address this, we have added a category for multi race.

Thank you for noticing our formatting inconsistency. We changed the format in Table 4 to match the format of the other tables in the manuscript, by center aligning the data.

  1. The meaning and impact of qualitative interviews from the participant in this survey must describe and discuss deeply, not only present the answers.

Response: We made the following changes to address your feedback:

(1) We provide further detail on our approach to analysis of qualitative data (Page 3, Lines 109-115): Two team members reviewed the first several transcripts independently and developed a preliminary coding matrix. The team members then manually coded all transcripts in full. Through constant comparison and iterative discussion, codes were revised and added until consensus on summary categories was reached. We finalized categories, grouped themes in each category, and identified quotes that best highlighted individual themes. Supporting statements are identified by participant number. 

(2) We formatted the data using bullets to outline the themes. For example (Page 5, Lines 145-155):

Pre-ban, 29% (4/14) participants strongly agreed or agreed with the Massachusetts menthol ban (Table 2), citing reasons that fell into three themes:

1) Ban would help individuals to stop smoking: “I will not be smoking them because I only smoke menthol. …Soon as June 1st come, I’m done. I’m not spending my money on something I don’t like” (P15)

2) Ban would benefit the community: “The young people smoke Newports. …[post-ban] they won’t be able to get ’em. I think that will help not only the young people, but the seniors as well as the middle age” (P25)

3) Ban would undo an injustice to the Black community: “It irritates me that I can’t buy a pack [of menthol cigarettes], but I know that they’ve been marketed, unfortunately, for people of color, so it’s possible that injustice will no longer be happening. It’s a good thing” (P26).

(3) To better clarify that we are using representative quotes to illustrate the themes from analysis of qualitative data, we added (Page 4, Lines 127-129): We present representative quotes, identified by participant number, that best illustrate themes captured from analysis of qualitative data. Perceived rationale for the menthol ban fell into three themes:

  1. The format of all the reference must follow the format of Int. J. Environ. Res. Public Health.

Response: We have now formatted the references to style format of Int. J. Environ. Res. Public Health.

We amended reference #10. It now cited as (Page 13, Lines 400-402): U.S. Food and Drug Administration. (2021, April 29.) FDA Commits to Evidence-Based Actions Aimed at Saving Lives and Preventing Future Generations of Smokers [Press release]. https://www.fda.gov/news-events/press-announcements/fda-commits-evidence-based-actions-aimed-saving-lives-and-preventing-future-generations-smokers.

Reviewer 3 Report

1. The statistical analysis is reported as basic descriptive, but is not depicted in the tables. It is not clear how and what kind of statistical analysis is being done on which samples, as it is not shown or discussed anywhere except the data analyses section. 

2. The manuscript does not include reference to any of the following articles: Asare S et al (PMID: 36848121; 36107431; 34982100)  or Pearson et al (PMID: 22994173). These articles should be provided as reference articles for the introduction or discussion.

Author Response

Reviewer: 3

Specific Comments to Author

  1. The statistical analysis is reported as basic descriptive, but is not depicted in the tables. It is not clear how and what kind of statistical analysis is being done on which samples, as it is not shown or discussed anywhere except the data analyses section. 

Response: Thank you for your feedback. We have added the following (Page 3, Line 101-106):

Pre-ban and 6-months post-ban questionnaire responses were summarized (frequency and percentages) in the Tables as follows: Table 1: Baseline characteristics of participants; Table 2: Participants’ perceptions of the ban on attitudes and menthol cigarette sales (pre- and 6-months post ban); Table 3: Perceived influence on smoking behavior (6-months post ban); and Table 4: Purchasing and use of tobacco products (pre-and 6-months post ban). In addition, paired t-tests were used to compare mean cigarettes smoked per day at the pre-ban versus 6-month post-ban timepoints.

  1. The manuscript does not include reference to any of the following articles: Asare S et al (PMID: 36848121; 36107431; 34982100) or Pearson et al (PMID: 22994173). These articles should be provided as reference articles for the introduction or discussion.

Response: Thank you for sharing these references with us. We agree that these articles should be included, and we apologize for the oversight. We have made the following revisions based on your recommendations:

 (1) We added (Page 11, Lines 286-288): In a cross-sectional survey to assess behavioral intentions of menthol smokers in the event of a national ban, nearly 40% of menthol smokers said they would quit smoking altogether if menthol tobacco products were no longer available.(23)

  1. Pearson, J.L.; Abrams, D.B.; Niaura, R.S.; Richardson, A.; Vallone, D.M. A ban on menthol cigarettes: impact on public opinion and smokers' intention to quit. Am J Public Health 2012, 102, e107-114, doi:10.2105/AJPH.2012.300804.

(2) The manuscript by Asare and colleagues was not yet published at the time of our manuscript submission, and we have since reviewed the manuscript. We have added (Page 11, Lines 301-305): A recent analysis of the nationally representative, cross-sectional Behavioral Risk Factor Surveillance System showed the MA ban to be associated with a 1% decline in current cigarette smoking rates in Massachusetts than comparison states. The study showed a greater reduction in current cigarette smoking among males and Hispanic females while increasing among Black females.(24)

  1. Asare, S.; Majmundar, A.; Xue, Z.; Jemal, A.; Nargis, N. Association of Comprehensive Menthol Flavor Ban With Current Cigarette Smoking in Massachusetts From 2017 to 2021. JAMA Intern Med 2023, doi:10.1001/jamainternmed.2022.6743.

Round 2

Reviewer 1 Report

I would like to thank the authors for being attentive to my previous feedback. I believe the manuscript is now suitable for publication.

Author Response

Thank you for the positive feedback.